# 2-Pyridylmetallocenes, Part IX. Sulphur-Substituted 2-Pyridylferrocene: Synthesis and Reactivity towards Pt(II) and Hg(II)

**DOI:** 10.3390/molecules29204884

**Published:** 2024-10-15

**Authors:** Stefan Weigand, Karlheinz Sünkel

**Affiliations:** Department of Chemistry, Ludwig-Maximilians University Munich, 81377 Munich, Germany

**Keywords:** pyridylferrocenes, cyclomercuration, platinum complexes, N,S-ligands

## Abstract

Thio-substituted 2-pyridylferrocenes [CpFe{C_5_H_3_(X)(C_5_H_4_N)}] (X = SOTol, **3**; SMe, **5**) were prepared from [CpFe(C_5_H_4_R)] (R = SOTol, **1**; 2-C_5_H_4_N, **2**) in moderate yields. The reactions of **3** and **5** with [PtCl_2_(DMSO)_2_] yielded the binuclear N, S chelated complexes [CpFe{C_5_H_3_(X)(C_5_H_4_N)}-(к-N,S)-PtCl_2_] (X= SOTol, **4**, SMe, **6**), while the reaction of **5** with Hg(OAc)_2_/LiCl led to cyclomercuration with generation of [CpFe{C_5_H_2_(SMe)(C_5_H_4_N)(HgCl)}], **7**. The crystal structures of **6**·CH_2_Cl_2_ and **7** were determined. The structure of **6** showed a weak intramolecular Fe…Pt interaction and several weak intermolecular interactions involving all Cl atoms. Weak intermolecular interactions between Hg and S atoms in the cyclomercurated **7** led to a tetrameric structure involving a Hg_2_S_2_ ring.

## 1. Introduction

Ever since the discovery of the usefulness of ferrocene-based chiral ligands in catalytic dia- and enantioselective catalysis by Hayashi et al. exactly 50 years ago [1], this compound class has been extensively studied and optimized for particular applications. Most of these chiral ferrocene derivatives contain the element of planar chirality, mostly combined with at least one chiral substituent. Thus, many P^P, P^N, P^O, P^S, N^N, N^O, and N^S bidentate donor substituted ferrocenes were developed [2]. We would like to focus here on the N^S type ligands. For the N-donor substituent, either amino-alkyl groups [3], or heterocycles [4], mostly oxazolines [5,6], thiazolines [7,8], or pyridine derivatives [9,10,11] were used. The S-donor component was either an alkyl or aryl thioether [5,6,7,8] or a sulfoxide [9,10,11]. As for dia- or enantioselective catalysis, an enantiopure ligand is essential, and a diastereoselective synthesis of these planar-chiral ferrocenes is highly desirable. For this purpose, either ortho-metalations or C-H activations have been considered [12,13]. Numerous examples exist with metalating agents like lithium alkyls or amides, while for C-H activations Co(II), Cu(I,II), Sc(III), Rh(I,III), Pd(II) and, to a much smaller extent, the 5d metals Ir(III) and Au were used. While Pt(II)-catalyzed C-H functionalizations are well-known [14,15], they have apparently not been used in this context. Similarly, while cyclomercurated ferrocenylamines and their ability to undergo further transmetallation reactions are well-known [16,17], they also have not been used for further functionalization. Our group has been working on the further functionalization, including cycloplatination [18,19] and cyclomercuration [20] of various ferrocenylpyridines. Here, we report our studies on the synthesis of planar-chiral ferrocenylpyridines containing additional sulfenyl and sulfinyl substituents and their reactivity towards [PtCl_2_(DMSO)_2_] and Hg(OAc)_2_/LiCl.

## 2. Results

### 2.1. Synthesis

#### 2.1.1. 2-Pyridylferrocenes with Sulphur Containing Substituents

First, we looked at the synthesis of a tolylsulfinyl-substituted pyridylferrocene. Following Jäkle’s synthesis of 2-(3,5-dimethyl-pyridinyl)-1-tolylsulfinylferrocene [9], we treated (S)-tolylsulfinylferrocene **1** with LDA and ZnCl_2_ followed by Negishi coupling with 2-bromopyridine to obtain the desired 2-(2-pyridyl)-1-p-tolylsulfinylferrocene **3** as an orange powder in ca. 60% yield (Figure 1).

NMR spectroscopic investigation (^1^H, ^13^C, see Appendix A) showed that this product contained an unidentified impurity, which could not be removed by chromatography. Careful examination of the NMR spectra showed that the impurity contained neither a pyridyl nor a tolyl group (no additional signals of significant intensity at δ > 8.5 ppm and around δ = 2.4 ± 0.1 ppm in the ^1^H NMR spectrum, no additional signals at δ > 145 ppm and around 22 ± 1 ppm in the ^13^C-NMR spectra). However, the presence of some “aromatic” signals (around δ ≈ 7.5 ppm in the ^1^H-NMR and δ ≈ 130 ppm in the ^13^C-NMR spectrum) suggests that a PPh_3_ containing compound might be present.

Next, we investigated the synthesis of a thioether derivative of pyridylferrocene. Following the introductions of halide or carbonyl substituents described earlier by us [19,21], treatment of 2-pyridylferrocene with n-BuLi and MeSSMe produced the desired racemic 1-methylthio-2-(2-pyridyl)-ferrocene **5** as an orange viscous oil in 68% yield (Figure 2).

Compound **5** could be characterized by ^1^H, ^13^C-NMR (see Appendix A) as well as by MS (FAB+) and elemental analysis. 

#### 2.1.2. Reaction of Compound **3** with [PtCl_2_(DMSO)_2_]

With the intention of inducing a cycloplatination reaction, we used the reaction conditions we had successfully applied earlier for the cycloplatinations of other substituted pyridylferrocenes [18,19], for the reaction of (impure) **3** with [PtCl_2_(DMSO)_2_]. However, instead of a C-H activation reaction at the ferrocene core, a simple N,S coordination at the substituents occurred (Figure 3).

The presumed structure **4** is based mainly on the NMR data (see Appendix A). The ^1^H-NMR and ^13^C spectra show mainly rather large amounts of ethyl acetate (from chromatography) as an impurity. There are some further weak peaks in the ^1^H-NMR spectrum (there are many “satellite” signals next to the big ones, due to ^1^H-^195^Pt coupling), which, however, have no counterparts in the ^13^C NMR spectrum, and must therefore correspond to rather small and insignificant amounts. 

Hoping to be more successful, we studied also the reactivity of the sulfenyl substituted compound **5** towards the same “cycloplatination” conditions. However, again no C-H activation occurred, but only a S,N–chelation by Pt (Figure 4). The chelate compound **6** could be isolated as an oil in 30% yield and be characterized by ^1^H and ^13^C NMR (see Appendix A) as well as by MS (DEI). After careful recrystallization, a few crystals of **6** could be isolated that were suitable for X-ray diffraction.

As we had successfully cyclomercurated several pyridylmetallocenes [20,22], we also decided to look at a possible cyclomercuration of **5**. To our delight, a C-H activation next to the pyridyl substituent occurred and the cyclomercurated product 7 could be isolated after chromatography and recrystallization in low yield (Figure 4). Despite the low yield, it was possible to obtain ^1^H and ^13^C NMR spectra (see Appendix A) as well as MS and crystals suitable for X-ray structure determination.

### 2.2. Crystallographic Studies

#### 2.2.1. Molecular Structures of Compounds **6** and **7**

Compound **6** crystallizes together with one molecule of dichloromethane as a twin in the trigonal space group *P*3_2_. Since this is a polar space group, the selected crystal contains only the *S*_p_ enantiomer of this planar-chiral molecule. An ortep3 view is shown in Figure 1, and the most important bond parameters are collected in Table 1.

The most obvious feature is the bending down of the PtCl_2_ unit out of the cyclopentadienyl plane towards the iron atom, which also affords a rotation of the pyridine ring out of this Cp plane, and also a slight “tilting” of the Cp-M-Cp unit. A search in the CSD (accessed on 22 September 2024) shows only four compounds that contain a ferrocene-based S,N ligand coordinated to a M′Cl_2_ fragment, all with M′ = Pd (ceycam, gendae, keyjed, seybaz). Two of them show the same arrangement with the coordinated fragment on the proximal side as the iron atom [7,23], while in the other two the MCl_2_ group is on the distal side. Still, the Pt…Fe distance of 3.822(3) Å is larger than the sum of van-der-Waals radii (by 0.102 Å), as is true also for the two Pd…C contacts in the mentioned [CpFe{C_5_H_3_(SMe)(C_3_H_4_NS)}-PdCl_2_] and [CpFe{C_5_H_3_(S*^t^*Bu)(CH_2_NMe_2_)}-PdCl_2_]. The bite angle of the SPtN chelate is 93.8(7)°, compared with 91.02(7) and 100.2(2)° in the two Pd compounds. Further geometrical parameters of the structure can be found in Table 1.

Compound **7** crystallizes in the triclinic space group *P*-1 with two molecules in the asymmetric unit. Figure 2 shows an ortep3 top view of both molecules. The chlorine atom of molecule B is disordered over two positions. As the space group is centrosymmetric, both enantiomers of this planar-chiral compound are present in the crystal. There are only minor differences between the two independent molecules (see Table 1). The pyridine rings are only slightly rotated out of the plane of the Cp rings, with the N atoms on the distal sides. In the structurally related [CpFe{C_5_H_2_(CH_3_)(C_5_H_4_N)(HgCl)}], the pyridyl ring is slightly rotated the other way, while in [CpFe{C_5_H_3_(C_5_H_4_N)(HgCl)}] it is nearly perfectly coplanar with the Cp ring [20]. The Hg…N distance in molecule A of compound **7** is significantly shorter than in molecule B, and it is also shorter than in the just mentioned literature compounds (2.74 and 2.71 Å, respectively) and other known cyclomercurated ferrocenylimines [17].

#### 2.2.2. Intermolecular Contacts and Packing Diagrams

Both compounds **6** and **7** show non-classical intra- and intermolecular hydrogen bonds. In compound **6**, only Cl atoms act as acceptors, both the Pt bound chlorines and the chlorines of the lattice dichloromethane (Figure 3). There are some H…H contacts and C-H…C contacts between pyridine CH bonds and a C-C bond of a neighboring C_5_H_5_ ring. In addition, the pyridine ring acts as an acceptor of two C-Cl…π bonds towards the CH_2_Cl_2_ molecule and a very weak π-π interaction with the unsubstituted Cp ring.

In compound **7**, chlorine as well as sulphur atoms act as acceptors (Figure 4). Table 2 collects the parameters of the H bonds of compounds **6** and **7**. Besides the H bonds, there are also some weak interactions involving Hg1 and S2 atoms, leading to a four-membered Hg_2_S_2_ ring formed by two molecules A and two molecules B each. (Figure 5). Quite interestingly, Hg2 and S1 do not take part in similar weak interactions.

The interplay of the different interactions led to the packing diagrams shown in Figure 6 and Figure 7.

## 3. Discussion

Both synthetic approaches, namely either lithiation of a sulphur containing ferrocene followed by introduction of the pyridinyl substituent, or lithiation of a pyridylferrocene followed by introduction of a sulphur electrophile, led to the synthesis of planar-chiral ferrocenyl N,S-ligands **3** and **5**. Unfortunately, these reactions were not 100% chemo-selective, and other products were also formed, which could neither be identified nor be fully separated, without large losses of yield. From our experience with lithiations of both tolylsulfinylferrocene **1** and 2-pyridylferrocene **2**, we know that side reactions like mono-lithiation in 3 position or dilithiations either in 2,3 or 2,1’ positions are possible; however, in earlier cases chromatographic separations were possible. Still, characterization by NMR methods, and, in part, by mass spectrometry and elemental analysis was possible; however, no single crystals of **3** or **5** could be obtained.

Reaction of iminoferrocenes with [PtCl_2_(DMSO)_2_] under basic conditions are known to yield cycloplatinated complexes with both Pt-C and Pt-N bonds. While there are apparently no examples of ferrocenyl N,S ligands in cycloplatination reactions—at least in the crystallographic literature, there are two examples of cycloplatinated iminobenzenes containing also coordinating S atoms (aboxiy and fugkuo, [24,25]). However, in both cases, the sulphur atom was not directly attached to the benzene ring. When, however, the SCH_3_ group was bound in ortho-position to the imino group, the Pt atom was only chelating the substituents in a N,S fashion (xeqquh [26]). Therefore, it was not surprising that both compounds **3** and **5** acted only as N,S chelate ligands towards PtCl_2_ producing the bimetallic complexes **4** and **6**. Both compounds could be characterized by NMR methods, and compound **6** also by X-ray diffraction. The crystal of **6** studied contained only the enantiomer with planar S configuration; however, since there were no chiral crystallization conditions, it is most likely that it was only the serendipitous picking of this particular crystal, and not a property of the whole solid product.

While no cycloplatination could be observed either with **3** or **5**, clean cyclomercuration occurred with the latter compound, reacting next to the pyridyl substituent only, giving **7**. Cyclomercuration of tolylsulfinylferrocene **1** has been reported to yield a compound, where the oxygen atom of the sulfinyl group was also coordinated to the Hg atom [27]. Although we did not study the analogous reaction of **3**, we assume that for steric reasons the mercuration would occur here also next to the pyridinyl substituent. We had already shown earlier that both 2-pyridylferrocene and 2-methyl-1-(2-pyridyl)ferrocene underwent cyclomercuration reaction, yielding products with a T-shaped primary coordination sphere. While the product of the mono-substituted pyridylferrocene had shown no further interactions involving the Hg atom, the methyl-substituted product did: Some kind of “dimerization” had occurred in the crystal, yielding four-membered Hg_2_Cl_2_ rings. Our new compound **7** showed a “tetramerization”, this time using the S atoms of neighboring molecules producing a four-membered Hg_2_S_2_ ring.

## 4. Materials and Methods

### 4.1. Starting Materials and Instrumentation

Reagents LDA solution (1.0 M in THF/hexane), LiEt_3_BH (1.0 M), *n*-butyl lithium (2.5 M), ZnCl_2_, NaOAc, Hg(OAc)_2_, LiCl, 2-bromopyridine, MeSSMe were obtained commercially and used as provided. Solvents (THF, Et_2_O, hexane, petroleum ether, dichloromethane, ethyl acetate) were obtained in analytical grade and were saturated with N_2_. Compounds **1** and **2** as well as [Pd(PPh_3_)_2_Cl_2_] and [PtCl_2_(DMSO)_2_] were prepared according to the literature.

NMR spectra were measured on jeol ecp-270 or ex-400 instrument, (both jeol Germany, Freising, Germany) using CD_2_Cl_2_ or CDCl_3_ as solvent. The chemical shifts were obtained relative to the residual solvent signals, as defined by the MestReNova software (Version 14.1.1-24751, Mestrelab Research S.L., Santiago de Compostela, Spain) (δ_CDCl3_ = 7.260 and 77.16 ppm, respectively; δ_CHDCl2_ = 5.320 and 53.84 ppm, respectively). Mass spectra were obtained on Finnigan MAT 90 and JEOL Mstation 700 (jeol Germany, Freising, Germany) instruments, in DEI or FAB mode. Chromatographic separations were performed in Schlenk glass frits of ca. 20 mm diameter with a filling hight of ca. 15 cm of silica gel, which had been suspended in petroleum ether. A slight overpressure of nitrogen was used to promote the migration of the bands through the column.

Crystal structure determinations were performed on an Oxford XCalibur3 diffractometer (Oxford Diffraction Ltd., Oxford, UK). Further details can be found in Appendix A.

### 4.2. Synthesis of 1-(S)-(P-Tolylsulfinyl)-2-(2-pyridyl)-ferrocene (**3**)

A solution of **1** (0.65 g, 2.00 mmol) in THF (20 mL) was treated at −78 °C with a 1.0 M solution of LDA in THF/hexane (2.20 mL, 2.20 mmol) with stirring for 1 h. Then, a solution of ZnCl_2_ (0.30 g, 2.20 mmol) in THF (10 mL) was added and the mixture was warmed to r.t. After stirring for another hour at this temperature, a 1.0 M solution of LiEt_3_BH (0.20 mL, 0.20 mmol) and solid [PdCl_2_(PPh_3_)_2_] (0.14 g, 0.20 mmol) were added, which led to formation of a black suspension. 2-Bromopyridine (0.20 mL, 2.00 mmol) was added and the suspension was stirred for 20 h. After addition of 1.0 m NaOH solution (20 mL), a two-phase mixture was obtained, which was extracted three times with CH_2_Cl_2_ (15 mL each). The combined organic extracts were washed with water and brine and then dried over MgSO_4_. After complete evaporation of the solvent in vacuo, the residue was taken up in the minimum amount of CH_2_Cl_2_ and chromatographed on alumina. A first, yellow fraction, was obtained with PE/EtOAc 4:1 and was discarded. A second yellow fraction eluted with PE/EtOAc 1:1 contained a mixture of unreacted starting material and product, and it was discarded also. Finally, a EtOAc/CH_2_Cl_2_ mixture was used as eluent, and after evaporation compound **3** was obtained as orange-yellow powder (0.52 g, 1.30 mmol, 65%).

^1^H-NMR: (CDCl_3_, 400 MHz): δ = 8.59 (m, 1H), 7.89 (m, 1H), 7.72 (m, 2H), {7.69–7.60 (m, 2H), 7.54 (m, 0.5H), unidentified impurity}, 7.46 (m, 1H), 7.29 (m, 2H), 7.13 (m, 1H), 5.09 (m, 1H), 4.47 (m, 1H), 4.16 (m, 1H), 4.14 (s, 5H), 2.41 (s, 3H).

^13^C-NMR (CDCl_3_, 101 MHz): δ = 156.8, 149.3, 141.3, 140.8, 136.2, 129.4, {132.3, 132.2, 128.7, 128.6; unidentified impurity}, 125.9, 123.0, 121.6, 93.3, 85.9, 71.7, 71.5, 71.4, 70.1, 21.6.

### 4.3. Reaction of 3 with [PtCl_2_(DMSO)_2_]

A solution of **3** (0.080 g, 0.20 mmol) and [PtCl_2_(DMSO)_2_] (0.090 g, 0.20 mmol) in toluene (40 mL) was treated with a solution of NaOAc·3H_2_O (0.030 g, 0.40 mmol) in MeOH (1 mL), and the mixture was heated to reflux for 3 h. After cooling down to r.t., the solvents were evaporated in vacuo, and the obtained red residue was chromatographed on alumina. A first fraction was obtained with PE/CH_2_Cl_2_, and it was discarded. A second fraction was obtained using EtOAc/CH_2_Cl_2_ 1:2 as eluent. Evaporation left a red solid (0.08 g), presumably [{Fe(C_5_H_5_)(C_5_H_3_(2-C_5_H_4_N)(SO-*p*-Tol))}-(*κ*-N,S)-PtCl_2_] (**4**) (0.12 mmol, 61%)

^1^H-NMR (CDCl_3_, 270 MHz): δ = 9.41 (m, 1H), 8.07 (m, 2H), 7.87 (m, 1H), 7.59 (m, 1H), 7.44 (m, 2H), 7.32 (m, 1H), 5.13 (m, 1H), 4.81 (s, 5H), 4.74 (m, 1H), 4.10 (m, 1H), 2.50 (s, 3H).

^13^C-NMR (CDCl_3_, 68 MHz): δ = 156.2, 155.3, 144.6, 139.3, 129.1, 128.7, 128.3, 124.6, 124.4, 89.0, 82.9, 74.2, 71.5, 69.4, 68.7, 21.7.

### 4.4. Synthesis of Racemic 1-Methylthio-2-(2-pyridyl) Ferrocene (**5**)

A solution of 2-pyridyl-ferrocene (**2**, 0.24 g, 0.91 mmol) in Et_2_O (10 mL) was treated with 2.5 m *n*-BuLi solution (1.0 mL, 2.50 mmol) at r.t. with stirring for 1 h. Then, the mixture was cooled to −78 °C and MeSSMe (0.27 mL, 3.0 mmol) was added. After stirring for 10 min at this temperature, the cooling bath was removed and stirring was continued for another hour. After evaporation of the solvent in vacuo, the obtained orange-brown residue was chromatographed on alumina and PE/CH_2_Cl_2_ 1:1 eluted the desired product [Fe(C_5_H_5_){C_5_H_3_(SMe)(C_5_H_4_N-2)-1,2}] (**5**), obtained as an orange-colored highly viscous oil (0.19g, 0.61 mmol, 68%).

^1^H-NMR (CDCl_3_, 270 MHz): *δ* = 8.54 (m, 1H), 8.17 (m, 1H), 7.63 (m, 1H), 7.11 (m, 1H) 4.97 (m, 1H), 4.49 (m, 1H), 4.37 (m, 1H), 4.09 (s, 5H), 2.27 (s, 3H).

^13^C{^1^H}-NMR (CDCl_3_, 68 MHz): *δ* = 158.5, 149.0, 135.7, 122.5, 121.0, 85.2, 83.3, 73.3, 71.1, 69.6, 68.6, 20.1.

MS (FAB^+^): *m*/*z* = 307.3 (M^+^), 263.3 (M^+^–SCH_2_).

EA (calc/found): C: 62.15/61.92; H: 4.89/4.97; N: 4.53/4.52; S: 10.37/10.85%.

### 4.5. Synthesis of [Fe(C_5_H_5_){C_5_H_3_(SCH_3_)(C_5_H_4_N-2)-1,2}]-(κ-N,S)-PtCl_2_] (**6**)

A solution of **5** (0.077 g, 0.25 mmol) and [PtCl_2_(DMSO)_2_] (0.11 g, 0.26 mmol) in toluene (50 mL) was treated with a solution of NaOAc·3H_2_O (0.068 g, 0.50 mmol) in MeOH (1.0 mL). After refluxing this mixture for 5 h and cooling down to r.t., the solvents were evaporated in vacuo. The residue was placed on top of an alumina column. CH_2_Cl_2_/EtOAc 2:1 eluted an orange band, which yielded after evaporation **6** (0.043 g, 0.075 mmol, 30%) as an orange-colored oil. Recrystallization from CH_2_Cl_2_ at r.t. gave a few single crystals, suitable for X-ray diffraction.

^1^H-NMR (CDCl_3_, 270 MHz): *δ* = 9.39 (m, 1H), 7.81 (m, 1H), 7.50 (m, 1H), 7.26 (m, 1H), 4.99 (m, 1H), 4.95 (m, 1H), 4.77 (s, 5H), 4.71 (m, 1H), 2.16 (s, 3H).

^13^C{^1^H}-NMR (CD_2_Cl_2_, 68 MHz): *δ* = 162.6, 156.6, 139.1, 125.7, 124.0, 74.0, 73.6, 71.9, 69.1, 28.2.

MS (DEI): *m*/*z* = 575.0 (M^+^)

### 4.6. Synthesis of [Fe(C_5_H_5_){C_5_H_2_(HgCl)(SCH_3_)(C_5_H_4_N-2)-1,3,2}-(κ-C,N)] (**7**)

A solution of Hg(OAc)_2_ (0.20 g, 0.63 mmol) in MeOH (25 mL) was added to a solution of **5** (0.19 g, 0.61 mmol) in CH_2_Cl_2_ (12 mL). After stirring for 3 h at r.t. a solution of LiCl (0.05 g, 1.2 mmol) in MeOH (19 mL) was added. After stirring for another three days a mixture of CH_2_Cl_2_ (15 mL) and water (20 mL) was added. The aqueous phase was separated and extracted with CH_2_Cl_2_ (40 mL). The combined organic phases were dried over MgSO_4_, and after filtering, the solvents were evaporated in vacuo. The obtained orange-colored residue was chromatographed on silicagel. A mixture PE/CH_2_Cl_2_ (gradient 2:1 → 1:2) eluted the title compound **7** as orange micro-crystals (0.090 g, 0.16 mmol, 27%). Pure CH_2_Cl_2_ eluted another orange fraction (0.080 g after evaporation), which contained **7** besides other unidentified compounds; further elution with CH_2_Cl_2_/EtOAc 3:1 yielded after evaporation apparently unreacted **5** (0.080 g, 42% recovery). Recrystallization of the first fraction from CH_2_Cl_2_ at r.t. yielded some crystals suitable for X-ray diffraction.

^1^H-NMR (CD_2_Cl_2_, 270 MHz): *δ* = 8.66 (m, 1H), 8.45 (m, 1H), 7.71 (m, 1H), 7.22 (m, 1H), 4.73 (m, 1H), 4.38 (m, 1H), 4.08 (s, 5H) [* actually this signal overlapped with the signal of an impurity, which is most likely derived from metalation of compound **2**], 2.34 (s, 3H).

^13^C{^1^H}-NMR (CD_2_Cl_2_, 68 MHz): *δ* = 158.4, 148.4, 137.6, 122.1, 121.9, 86.6. 86.5, 83.9, 79.0, 75.5, 71.9, 21.0.

MS (DEI^+^). *m*/*z* = 545.3 (M^+^) {among others, in F2: 591.3 (M^+^, **8**)}.

## Data Availability

Copies of all NMR spectra relevant to this study are collected in the Appendix A. The experimental details of the crystal structure determinations can be found in Appendix A. The crystallographic cif files have been deposited with the Cambridge Structural database. CCDC 2386172–2386173 contain the supplementary crystallographic data for this paper. These data can be obtained free of charge via www.ccdc.cam.ac.uk/data_request/cif, by emailing data_request@ccdc.cam.ac.uk, or by contacting The Cambridge Crystallographic Data Centre, 12, Union Road, Cambridge CB2 1EZ, UK; fax: +44 1223336033.

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
