# Peer review of "2-Pyridylmetallocenes, Part IX. Sulphur-Substituted 2-Pyridylferrocene: Synthesis and Reactivity towards Pt(II) and Hg(II)"

_molecules, 2024, doi:10.3390/molecules29204884_

Round 1

Reviewer 1 Report

Comments and Suggestions for Authors

This manuscript by Weigand and Sünkel describes the synthesis of two novel thio-substituted 2-pyridylferrocenes and their reactivity towards PtCl2(DMSO)2 and Hg(OAc)2/LiCl. Interestingly, reactions with pt(II) give rise to chelating N^S Pt complexes while with Hg(II) the reaction leads to a cyclomercuration product. All new compounds have been characterized using standard methods, including single crystal X-ray diffraction analysis that shows a tetrameric structure involving a Hg2S2 ring in the case of the cyclomercurated derivative. The work is competently carried out and the manuscript is well written up, presented and organized. Overall, the general quality and scientific reliability of this paper is high and it deserves to be published in MOLECULES as it is.

Very minor issues:

l. 54: ris-substituted  should be bis-substituted

l. 108: C3H4N should be C5H4N (in the first formula)

l. 206: reference(s) must be called after «literature»

l. 294: the signal at 4.08 should integrate for 5H, or some signals are lacking.

Ref. [26] abbreviate Transactions to Trans.

It is not necessary to write a long report on a short paper describing a work perfectly carried out. The strong and long experience of the authors - especially Prof. Sünkel - in this field of organometallic/metallocene chemistry is by itself a guarantee of scientific quality. As stated in my "short" report, the originality of this paper lies in the different reactivity of two M(II)Cl2 toward the same ferrocenyl substrate, leading to chelation vs. Cyclometallation. Adequate methodology has been used and no improvement is needed. The conclusions are consistent with the evidence and arguments presented, addressing the main question posed in the Introduction. In addition,the manuscript is appropriately referenced. No doubt that this manuscript will be an added-value item to the special issue to which it has been submitted to.

Reviewer 2 Report

Comments and Suggestions for Authors

The paper entitled "2-Pyridylmetallocenes, Part IX. Sulphur-substituted 2-pyridylferrocene: Synthesis and reactivity towards Pt(II) and Hg(II)" contains valuable pieces of information about sulfur-substituted (sulfide and sulfoxide-group containing) ferrocene compounds and their Pt and cyclo-Hg complexes. 

The paper is well written, and the scientific value of the complexes is high. One point is questionable, what is the contaminant in the sulfide compound ?  Since I was not given the ESI, I could not check the NMR spectra. 

However,  I strongly suggest thinking about the followings - and insert some further words into the paper about this contaminant, if the authors can find some solution/can disclose some possibility about that according to the next possibilities.  

The bis-substituted compounds properties probably differ enough from the mono-substituted ones to be able to separate these from each other chromatographically, if the appropriate column/eluent are used. 

The presence of an isomer is possible, as the authors said, but these isomers could be distinguished by a detailed 13C NMR methods with checking the neighbour atom interactions effect on these spectra (1,2 or 1,3  isomers on Cp-ring).

Without checking these spectra, I cannot give direct suggestions, but please check the possibility that diastereomers are present (due to the pro-chirality of sulfoxyl group and planar chirality of ferrocene compounds). Basically, there is no optical induction to form diastereomers, but because the pyridylation is not complete, kinetic resolution (one enantiomer/diastereomer of the rac starting material reacts slower/faster than the other) may results in this phenomenon.    

Please check the spectra to disclose/confirm the possibility of these situations. 

There are some places where the "Sulfur" is wriitten with capital letters within sentences, these should be non-capital letter ("s"). 
